# An In Vivo Platform for Rebuilding Functional Neocortical Tissue

**DOI:** 10.3390/bioengineering10020263

**Published:** 2023-02-16

**Authors:** Alexandra Quezada, Claire Ward, Edward R. Bader, Pavlo Zolotavin, Esra Altun, Sarah Hong, Nathaniel J. Killian, Chong Xie, Renata Batista-Brito, Jean M. Hébert

**Affiliations:** 1Department of Neuroscience, Albert Einstein College of Medicine, Bronx, NY 10461, USA; 2Stem Cell Institute, Albert Einstein College of Medicine, Bronx, NY 10461, USA; 3Department of Neurological Surgery, Albert Einstein College of Medicine, Bronx, NY 10461, USA; 4Department of Electrical and Computer Engineering, Rice University, Houston, TX 77005, USA; 5Department of Genetics, Albert Einstein College of Medicine, Bronx, NY 10461, USA; 6Department of Psychiatry and Behavioral Sciences, Albert Einstein College of Medicine, Bronx, NY 10461, USA

**Keywords:** neocortex, transplant, tissue replacement, vascularization, layering

## Abstract

Recent progress in cortical stem cell transplantation has demonstrated its potential to repair the brain. However, current transplant models have yet to demonstrate that the circuitry of transplant-derived neurons can encode useful function to the host. This is likely due to missing cell types within the grafts, abnormal proportions of cell types, abnormal cytoarchitecture, and inefficient vascularization. Here, we devised a transplant platform for testing neocortical tissue prototypes. Dissociated mouse embryonic telencephalic cells in a liquid scaffold were transplanted into aspiration-lesioned adult mouse cortices. The donor neuronal precursors differentiated into upper and deep layer neurons that exhibited synaptic puncta, projected outside of the graft to appropriate brain areas, became electrophysiologically active within one month post-transplant, and responded to visual stimuli. Interneurons and oligodendrocytes were present at normal densities in grafts. Grafts became fully vascularized by one week post-transplant and vessels in grafts were perfused with blood. With this paradigm, we could also organize cells into layers. Overall, we have provided proof of a concept for an in vivo platform that can be used for developing and testing neocortical-like tissue prototypes.

## 1. Introduction

The transplantation of pluripotent stem cell-derived neural precursors into the cortex is an exciting potential approach to repair the brain. To achieve this goal, grafted cells must re-establish damaged neural circuits that participate in the restoration of lost behavioral function. Significant progress has been made in demonstrating the feasibility of transplanting precursor cells to replace neurons in the cortex. Graft-derived neurons can survive for years in mice [1] and differentiate into appropriate neuronal subtypes that exhibit normal electrophysiological activity, project long distances outside of the graft to appropriate targets, synaptically integrate with surrounding host neurons, and respond to sensory input and participate in motor output [2,3,4,5,6,7].

Despite these significant discoveries, it is unclear whether grafted neurons in the neocortex can encode useful behavior as a result of their electrophysiological activity. Reported behavioral benefits are instead a result of activity-independent functions such as the secretion of anti-inflammatory or neurotrophic factors [8,9]. The inability to demonstrate that electrophysiological activity of grafted neurons encode useful behavior is not surprising considering there are cortical cell types that are thus far missing in grafts, in addition to these grafts lacking normal cortical cytoarchitecture. While cerebral organoids display a subset of similar characteristics to a normal fetal cortex, their differentiation has thus far been abnormal after transplantation [10,11]. Therefore, there is currently no method of generating facsimiles of neocortical tissue in adults, whether for the purpose of study or therapy. 

The goal of this study is to provide an initial proof of concept for a neocortical grafting platform that supports (1) the survival and differentiation of the major neocortical cell types, (2) vascularization, (3) neuronal integration, and (4) layering. Toward this goal, we tested whether grafting cells in a three-dimensional scaffold could sustain the differentiation of all the major cortical cell types, vascularization, and a layered cytoarchitecture. Using dissociated mouse cortical fetal cells mixed with a commercial scaffold, we found that the neuronal, glial, and vascular components within the graft survived and successfully integrated with the host tissue. Our results suggest that this platform is suitable for future optimization and testing of structured, vascularized, multi-cell type neocortical tissue prototypes.

## 2. Methods

### 2.1. Animals

Animal experiments were approved by the Albert Einstein College of Medicine Institutional Animal Care and Use Committee in accordance with National Institutes of Health guidelines. Swiss Webster mice (Charles River), male and female, ages 2–6 months were used as hosts for this study. Donor embryos were harvested from transgenic mice appropriate for the experiment. For visualizing neural components, donors were from *Foxg1^cre^*^/+^ (Jax # 06084) mice crossed with homozygous *Rosa26^CAG-loxSTOPlox-eGFP^* (Jax #010701) mice. For visualizing graft-derived vascularization, donors were from Mesp1^cre/+^ mice [12] (donated from Paul Frenette’s lab, Albert Einstein College of Medicine) crossed with homozygous Rosa26^loxSTOPlox-tdTomato^ mice (Jax # 007909). For visualizing graft layers, donors were from *Foxg1^cre/+^* mice crossed with homozygous *Rosa26^CAG-loxSTOPlox-eGFP^* mice, and donors from *Foxg1^cre/+^* mice crossed with homozygous Rosa26^loxSTOPlox-tdTomato^ mice. For graft projections, donors were from *Foxg1^cre/+^* mice crossed with homozygous *Rosa26^CAG-loxSTOPlox-Chr2-EYFP^* (Jax # 012569) mice, or *Foxg1^cre/+^* mice crossed with homozygous Rosa26^loxSTOPlox-tdTomato^. Mice were housed on a 12 h light–dark cycle and provided with water and chow ad libitum. The number of mice used in each experiment and the total number of mice used in this study are listed in Table 1 and Table 2.

### 2.2. Fetal Telencephalon Dissociation

Donor cells were harvested and dissociated from E12.5 telencephalons with genotypes suitable for each experiment (see “Animals” above). Dissociation was performed with Accutase (Innovative Cell Technologies, San Diego, CA, USA, cat. AT104) followed by trituration of the cell suspension. Cells were washed, counted, and then resuspended at ~500 k cells per μL in mouse embryonic cell media (Neurobasal, B27, N2, pen/strep) diluted with Matrigel (Corning, New York, NY, USA, cat. 356234), with the addition of Methylprednisolone (8.7 mM; Sigma, MO, USA, cat. M3781) and VEGF (20 ng/mL; Sigma cat. GF445). 

### 2.3. Transplantation Procedures

Mice were placed under 5% isoflurane infused oxygen anesthesia and once anesthetized maintained at 2% isoflurane/oxygen. Mice were placed on a stereotaxic platform and head fixed with ear bars. The scalp was wiped with betadine prior to making a 6 mm incision over the midline. On one side of the neocortical hemisphere, a 2 mm diameter craniotomy was performed over the somatosensory cortex (centered at Bregma: A/P~1.0, M/L~2.0) followed by an incision in the dura with a scalpel. The dura was carefully peeled back and a biopsy punch was inserted 1 mm deep into the cortex. Any tissue within the cut area that was not extracted with the biopsy punch was removed with a blunt tip needle attached to a vacuum to create a uniform cylindrical lesion. The blood was flushed with saline until the bleeding stopped. The cell/Matrigel solution was then added to fill the lesion. The volume of the 1.25 mm diameter × 1 mm deep cylindrical lesion was 1.23 μL. Given that the cell suspension was at 500 K cells/μL, this meant a minimum of ~600 K cells were injected to fill the lesion. However, we routinely found it necessary to top off the solution with another 0.5 μL due to absorbance of fluid into neighboring tissue, giving a total of roughly ~750 K cells per graft site. The number of cells in layered grafts were similar (~375 K cells for each layer). Once the cell solution had gelled, a 3 mm cover glass window was superglued to seal the craniotomy. The scalp was sutured shut and the mouse was given subcutaneous injections of Flunixin (FlunixiJect, Henry Schein, New York, NY, USA) and Ceftriaxone (Pfizer, Andover, MA, USA) and monitored closely for 24 h followed by daily monitoring for signs of infection or morbidity. In cases where layering of cells was performed, the procedure was similar except GFP+ cells were added to fill half of the lesion (about 0.75 μL) to generate layer A. Once layer A had gelled (approximately 20–30 min.), tdT+ cells were added to the lesion on top of the GFP+ cells to form layer B. 

### 2.4. Tissue Processing and Immunohistochemistry

Mice were anesthetized and then perfused with saline followed by 4% paraformaldehyde in PBS (PFA/PBS), via IV cardiac puncture. Brains were removed, fixed overnight in PFA/PBS, placed in 30% sucrose for 1–2 days, frozen in OCT, and cryosectioned at 30 μm. Sections were incubated in blocking buffer (10% normal goat serum in 0.3% triton in PBS) for 1 h, incubated with primary antibodies (see antibody list in Table 2) overnight at 4 °C, washed with PBS, incubated with secondary antibody (1:600) and Hoescht (1:10,000) for 1–2 h at RT, washed, mounted with Flouromount G (ThermoFisher, New Jersey, United States, cat. 00-4958-02), and topped with a glass cover. Stained sections were imaged with an epifluorescent microscope.

### 2.5. Image Analysis

To quantify cortical markers, interneurons, oligodendrocyte lineage cells, and microglia in the graft, cells from 3 sections per mouse were counted by hand in Photoshop. For the contralateral cortex, the entire dorsal/ventral length from the pia to the corpus callosum was selected for quantification to account for variation in density between layers. To quantify synapses, SYN+ puncta were counted by hand. The software Angiotool was used to analyze the blood vessels [13]. The flattened Z-stacks of graft vasculature were generated with image J [14]. To quantify the number of vessels that are being perfused with blood, IB4+ vessels were divided by the total amount of vessels. 

### 2.6. In Vivo Live Imaging

Mice were imaged with a two-photon laser scanning microscope (Thorlabs Bergamo) using a femtosecond-pulsed laser (Chameleon Ultra II, Coherent) tuned to 910 nm or a 1055 nm femtosecond-pulsed laser (Fidelity 2, Coherent) and a 16X water immersion objective (0.8NA, Nikon, New York, NY, USA). ThorImage software was used to acquire the images. The mice were anesthetized with 1–2% isoflurane and head fixed. To visualize blood vessel structure, Z-stacks starting from the dorsal side of the graft were taken. Z-stacks were then flattened among the Z-axis with average intensity and processed with Angiotool. 

### 2.7. Perfusion of Blood Vessels

To visualize perfusion, mice were injected intravenously with labeled RBCs. Briefly, after harvesting blood from mice, RBCs were segregated from plasma and serum using centrifugation (500 g, 5 min), washed with saline, incubated for 1 hr at 37 °C with DiI (1:50) and DiO (1:50) (ThermoFisher cat. V2285; V2286), washed, and resuspended at 50% hematocrit. 100 μL of DiO/DiI-labeled RBCs were injected through the retro-orbital sinus while mice were under anesthesia. Circulating labeled RBCs could be seen in the vessels up to 3 weeks post-injection by 2p imaging. To label blood vessels post-hoc, 100 μL of IB4-647 was injected retro-orbitally 30 min before euthanasia. 

### 2.8. Headpost and Chronic Electrode Implantation Surgery 

On the day of the surgery, the mouse was anesthetized with isoflurane and the scalp was shaved and cleaned three times with Betadine solution. An incision was performed at the midline and the scalp resected to each side to leave an open area of skull. A headpost was glued with dental cement to the skull. Two 3 mm craniotomies were performed on both hemispheres centered at Bregma: A/P-2.7, M/L 2.5. On one hemisphere, a lesion was performed in V1 followed by the transplantation of donor cells from Foxg1^cre/+^;Rosa26^CAG-loxSTOPlox-eGFP/+^ mice. Mice were implanted with 2 probes: one in the graft and one in contralateral intact V1. Probes had 32 channels either in a linear or tetrode arrangement. Once probes were inserted, a 3 mm glass window was applied to each craniotomy then sealed with Kwik-Sil followed by super glue. One skull screw (McMaster–Carr) was placed at posterior pole. The skin was then glued to the edge of the Metabond with cyanoacrylate. Metabond was applied to secure the probes on the head. Analgesics were given immediately after the surgery and on the two following days to aid recovery. Mice were given a course of antibiotics (Sulfatrim, Butler Schein, Morriston, FL, USA) to prevent infection and were allowed to recover for 3–5 days following implant surgery before beginning spontaneous freely moving recordings. Mice were handled for 10 min/day for 5 days prior to the headpost surgery. After recovering from implant surgery, mice were habituated to treadmill fixation for 30–60 min per day for a total of 5 days.

### 2.9. In Vivo Electrophysiology

All extracellular single-unit and LFP recordings were performed with chronic ultra-flexible neural probes designed for eliciting minimal inflammation and for long-term recordings [15,16]. Signals were digitized and recorded by the RHD Recording System (Intan, Los Angeles, CA, USA). Data were sampled at 30 kHz for freely moving sessions and 20kHz for head fixed visual stimulus sessions. Recordings were performed during the light portion of the light/dark cycle.

### 2.10. In Vivo Freely-Moving Electrophysiology Recordings

Mice were placed in a clear, plastic box then connected to the RHD Recording System (Intan, Los Angeles, CA, USA) for 30 min, three times a week. 

### 2.11. Visual Stimulation

Mice were head fixed on a stereotaxic rig with a treadmill attached that allowed for movement of the limbs. Visual stimuli were presented on an LCD monitor at a spatial resolution of 1080 × 1080 pixels/cm, with a real-time frame rate of 60 Hz, and a mean luminance of 50 cd/m^2^ positioned 15 cm from the eye. The LCD monitors used for visual stimulation were positioned on both sides of the animal, perpendicular to the surface of each eye. Animals underwent two recording sessions per day, with monocular stimulation alternating between sessions. Both control and graft sides were recorded during the sessions regardless of the eye that was stimulated. Each session contained a 20 min block of gray screen followed by one hundred percent contrast drifting gratings in 12 orientations with a spatial frequency of 0.2 cycles/cm and temporal frequency of 2 cycles/second. Each stimulus was presented for 1 s followed by a 1.5 s interstimulus interval with 0.5 s jitter.

### 2.12. In Vivo Electrophysiology Analysis

Spikes were clustered semi-automatically using the following procedure. The Matlab package Kilosort2 was used to identify clusters through template matching [17]. Fast and accurate spike sorting of high-channel count probes was performed with Kilosort. Well-isolated units were curated using the Python library graphical user interface phy 2.0 [18]. Clusters contained no more than 10% auto correlogram contamination with a refractory period of 2 ms. Unit data were analyzed for firing rates and visual responses using custom-written Python code.

### 2.13. Single Unit Activity Analysis in Head Fixed Recordings

Spike waveforms were extracted and averaged from the high-pass filtered signal around the time of peak unit amplitude detected by Kilosort2. Firing rate was computed by dividing the total number of spikes a cell fired in a specified period by the total duration of that period. Baseline firing rate for individual neurons was computed as the average firing rate during the 500 ms before stimulus onset across trials. Stimulus-evoked firing rate for individual neurons was computed as the average firing rate during the 400 ms period after the onset of the visual stimulus across trials. Signal to noise ratio (SNR) was computed as the absolute value of the stimulus-evoked firing rate minus baseline firing rate divided by the sum of the baseline firing rate and the stimulus-evoked firing rate: SNR = abs (FR_evoked − FR_baseline)/(FR_evoked − FR_baseline). Orientation selectivity index was calculated as one minus the circular variance as described by Batista-Brito et al. [19]. Here, an OSI = 1 indicates that the cell only fires for a single stimulus orientation, whereas an OSI = 0 indicates no firing for that orientation. 

### 2.14. Local Field Potential Computation in Freely-Moving Recordings

Raw electrophysiology data were processed using SciPy signal processing toolbox. A low-pass Butterworth filter was applied at 200 Hz and subsequently downsampled to 1250 Hz. LFP power spectra were computed using Welch’s method, with a 500 ms Hanning taper and normalized over the total estimated power of the signal. Spectra from control and graft sides were averaged across 3–4 channels in a single animal for each recording time point.

### 2.15. Statistical Analyses

Statistical analyses were performed using GraphPad Prism. Data were analyzed for statistical significance using unpaired t test for the quantification of interneurons, oligodendrocytes, microglia, synapses, and blood vessels in the graft compared with the contralateral control. Multiple unpaired t test was used for quantitation of cortical layer subtypes. One-way ANOVA was used for quantitation of experiments examining Matrigel dilution. The *p* values and sample sizes for each experiment are listed in the results section and figure legends.

## 3. Results

### 3.1. Graft Integrity Is Dependent on Scaffold Dilution

To generate consistently sized lesions, cortical tissue was removed in one hemisphere of the cortex with a biopsy punch followed by aspiration to create a cylindrical cavity that reached the corpus callosum. Biopsy punches of 1.00, 1.25, and 2.0 mm in diameter resulted in reproducible lesions (Appendix A). For consistency, we used 1.25 mm lesions for this study unless otherwise indicated.

To deposit and maintain cells within the lesion sites, Matrigel was chosen as the scaffold due to its suitability for neuronal and vascular differentiation and survival [20,21,22]. To determine the optimal Matrigel dilution for transplanting mouse fetal cells, we compared Matrigel:medium dilutions of 1:2, 1:3, 1:4, and 1:5 (Appendix A). Dilutions of 1:2 and 1:3 led to inconsistently sized grafts, often with gaps between the scaffold and the host tissue, while 1:4 diluted grafts were more consistent in size and appeared to have a more uniform donor cell distribution. The area and perimeter of grafts at 1:4 dilution (N = 7) were significantly larger than grafts with 1:3 dilution (N = 6, *p* = 0.03 for area, 0.03 for perimeter) and 1:5 dilutions (N = 6, *p* = 0.04 for area, 0.02 for perimeter) (Appendix A). Although this study was performed with 1.25 mm diameter lesions, transplants into 2 mm lesions showed that the 1:4 dilution also evenly fills larger cylindrical lesions with cells that are uniformly distributed (Appendix A).

The density of blood vessels (labeled as CD31+/CD105+) within the grafts was also affected by the dilution of Matrigel (Appendix A). Grafts with a dilution of 1:3 (N = 3, *p* = 0.02) and 1:4 (N = 3, *p* = 0.002) had a lower vessel density while grafts with a 1:5 dilution (N = 4, *p* = 0.13) were not statistically different to the density in the intact contralateral cortex. This was not surprising because lower concentrations of Matrigel result in decreased matrix stiffness, allowing for more 3D structures such as vessels to form. Given the overall performance of the 1:4 Matrigel:media dilution, the remaining grafts were performed using this dilution.

### 3.2. Donor Neural Precursor Cells Survive and Differentiate into Several Cortical Cell Types 

To begin assessing whether the lesion and scaffold combination supports donor cell engraftment, we transplanted dissociated mouse cortical cells harvested from embryonic day (E) 12.5 telencephalon. The E12.5 telencephalon contains the cell types needed for proper development, such as neuronal, vascular, and glial precursor cells, and therefore is a practical source of cells for a proof of concept for this transplant model (Figure 1A) [23]. In addition, we previously observed that E12.5 cortex-derived vascular, glial, and neuronal precursor cells survive, differentiate, and integrate with the adult mouse cortex [24]. To trace donor cells, we harvested E12.5 telencephalon from *Foxg1^cre/+^*; *Rosa26^CAG-loxSTOPlox-eGFP/+^* mice, in which telencephalic cells are specifically labeled with GFP. Immediately after tissue dissociation, the donor cells were resuspended in the Matrigel:media solution with the addition of methylprednisolone, a corticosteroid known to decrease acute inflammation specifically in the context of brain transplantations [25]. Once the lesion was generated (in only one hemisphere of the neocortex), the site was rinsed with cold PBS until any bleeding ceased (usually 5–20 min). The cell/scaffold solution was then used to fill the lesion site, and the solution was gelled within 10–30 min. At 2 weeks post-transplantation (wpt), we assessed the cell type composition of the grafts and found that GFP+ cell bodies filled the entire space in the lesion and were not observed outside of the graft site (Figure 1B).

The neuronal donor cells differentiated into upper layer SATB2+ and deep layer CTIP2+ neurons (Figure 1C). The ratio of upper to deep layer neurons is crucial for function [26] and SATB2+ cells are normally more abundant (Figure 1D). We observed that at 2 wpt an abnormally low proportion of SATB2+ cells were present in the graft compared to CTIP2+ (SATB2: graft 57% of total SATB2 + CTIP2 + cells, N = 3; contralateral cortex 77%, N = 3, *p* = 0.003; CTIP2: graft 43%, N = 3; contralateral cortex 23% *p* = 0.003) (Figure 1D). The neurons in the graft also expressed synaptophysin (Figure 1E–F), a transmembrane protein that is part of synaptic vesicles often used as a marker for presynaptic terminals [27]. The density of synaptophysin puncta in the graft was lower than the contralateral density (graft 67,610 SYN+ puncta/mm^2^, N = 3; contralateral cortex 98,866 SYN+ puncta/mm^2^, N = 3, *p* = 0.03).

The cortex, among other brain regions, requires a balance of excitatory and inhibitory neurons in order to maintain and modulate information flow [28]. Interneurons (GABA+) and excitatory neurons (GABA-/NeuN+) were both present in the graft at levels comparable to the contralateral cortex (graft: 7% GABA+ out of total NeuN+ cells, N = 3; contralateral cortex, 9% GABA+, N = 3, *p* = 0.45) (Figure 1G,H). 

Glial cells including oligodendrocytes, astrocytes, and microglia form a large part of the brain and are required for brain development, maintenance, and function [29]. Mature oligodendrocytes produce myelin that insulates axons to facilitate rapid signal transmission. Cells in the oligodendrocyte lineage (OLIG2+) were present in the grafts at a similar density to the contralateral cortex (graft 9.7%, N = 3, contralateral cortex 6.3% N = 3, *p* = 0.2; Figure 1J,K). However, staining for myelin basic protein (MBP) revealed that grafts at 2 wpt were largely unmyelinated (Figure 1I). Some of the OLIG2+ cells had thin MBP+ processes extending from the cell body (Figure 1L), suggesting that the oligodendrocytes in the grafts were immature and might produce myelin at later timepoints.

Astrocytes are crucial for neuronal maturation and function and are an essential part of the neurovascular unit which forms the blood brain barrier (BBB) [30,31]. GFAP is a marker for astrocytes, but it is not normally detected in quiescent astrocytes in the neocortical parenchyma above the corpus callosum [32]. Nevertheless, GFAP+ astrocytes were detectable in a few peripheral areas of the graft (Figure 1M), suggesting lingering inflammation. However, given the unevenness and sparsity of these areas, GFAP+ cells were not quantified.

Microglia are essential to central nervous system’s development and homeostasis, playing roles in processes such as synaptic pruning, brain vascularization, and BBB maintenance [33,34,35]. Therefore, microglia are likely vital to the maturation and function of grafted cortical-like tissue. As expected, microglia (IBA1+) were present in the grafts at 2 wpt (Figure 1N-0), albeit at a lower density than contralateral cortex (graft 40.8 cells/mm^2^ N = 3, contralateral cortex 91.1/mm^2,^
*p* = 0.21). Microglia migrate into the mouse cortex from E9.5 until E14.5, and are therefore present in our donor cell mix [24]. However, because microglia do not express *Foxg1* and are, thus, not labeled with GFP, microglia in the graft could be donor- or host-derived.

### 3.3. Transplants Become Vascularized with Vessels and Perfused with Blood

To determine whether the grafts become vascularized using the same experimental setup as in Figure 1A, we co-immunostained grafts with the vascular endothelial markers CD31 and CD105. At 2 wpt, vessels appear present throughout the grafts (Figure 2A,B). Vascularization is a dynamic process that occurs in response to hypoxic cues in the environment; therefore, we examined the graft vessels at a later time point [36]. The vessels in the graft appeared to still be changing their morphology until at least 4 wpt (Figure 2C–E). From 2 to 4 wpt, despite having similar densities (2 wpt 12.2% vessels % area, N = 6; 4 wpt 8.3% vessel % area, N = 3, *p* = 0.1) and numbers of junctions (2 wpt 79/mm^2^, N = 6; 4 wpt 30.1/ mm^2^, N = 3, *p* = 0.08), the grafts exhibited significantly increased lacunarity at 4 wpt, a measure of irregularity or gaps within the network that does not rely on vessel density or caliber [37] (2 wpt 0.72, N = 6; 4 wpt 1.1, N = 3, *p* = 0.02). Consistent with ongoing maturation of graft vasculature over time, flattened Z-stacks acquired from 2-photon live imaging at 1, 2, and 4 wpt suggested that vessel length also increased between 1 and 4 wpt (Figure 2I,J).

Since the graft vasculature appeared to still be changing within 4 wpt, we compared the graft vasculature at 4 wpt to host vasculature in the contralateral hemisphere (Figure 2F–H). The contralateral vessels were denser than those within the graft (graft 8.3 vessel % area, N = 3; contralateral cortex 18.2 vessels % area, N = 5, *p* = 0.0004) and had a greater density of junctions (graft 30.6/ mm^2^, N = 3; contralateral cortex 122.4%, N = 5, *p* = 0.02). However, the vasculature of the graft was significantly more irregular than the contralateral vasculature (graft lacunarity = 1.1, N =3; contralateral cortex 0.3, N = 5, *p* < 0.0001). 

Vascular endothelial cells are present in mouse telencephalon at E12.5 and, therefore, present in the donor cell population [24]. To determine the source of the donor versus the host vessels, we transplanted donor cells from Mesp1^cre/+^; Rosa26^loxSTOPlox-tdTomato/+^ embryos to visualize the donor vessel contribution. At 4 wpt, there were very few donor vessels found in the graft. 

To test whether the graft vessels perfused blood, we intravenously injected red blood cells (RBCs) labeled with DiO and DiI and imaged blood vessel perfusion in real time with 2-photon microscopy. Blood vessel perfusion with labeled RBCs was observed as early as 7 days post-transplant (Appendix A). All of the grafts (5/5) that were imaged had vessels and RBCs circulating through them. To quantify how many of the graft vessels were perfused with blood, the mice were given intravenous injections of isolectin-B4 conjugated to Alexa Fluor 647 to label vessels through circulating blood (Figure 2J–L). Most of the vessels in the graft were labeled with IB4-647 at 30 dpt (78.5 % IB4+ vessels, N = 3). 

### 3.4. Donor Cells Can Be Layered in Lesion Sites

Lamination is a canonical feature of the neocortex throughout development and in its mature state. For example, during mid corticogenesis, the cortex is layered into a ventricular zone, subventricular zone, subplate, cortical plate, and marginal zone. Thus, to better recapitulate the developing cortex, a transplant model should allow for the layering of cells. To test whether donor cells can be layered in a lesion by sequential gelling of scaffold-cell mixes, donor cells labeled with GFP were manually deposited in the lower half of the lesion, allowed to gel, and overlaid with donor cells labeled with tdTomato to fill the remaining space in the lesion (Figure 3A). At 2 wpt, the donor cells were still organized in two layers, as originally grafted (Figure 3B). The donor cells were not seen outside of their respective layer. Despite the distinct border between layers, cellular structures such as neural processes and blood vessels were still able to cross between the layers (Figure 3C,D).

### 3.5. Donor Neurons Project to Appropriate Targets in Host Brains

To be useful, a transplant model must also allow axons of donor neurons to project outside of the graft to appropriate brain targets in the host. To facilitate visualization of the donor cell processes, we transplanted cortical fetal cells that were expressing either channelrhodopsin-YFP or tdTomato into a lesion in the somatosensory cortex. At 2 wpt, the donor neurons were observed to project to several structures in the host brain (N = 7/9 mice) (Figure 4A). Large numbers of fluorescent processes were seen projecting from the graft through the corpus callosum to the contralateral cortex (N = 6/9 mice) (Figure 4(1,3,4)) [38]. Many fibers were present in the adjacent cortex (N = 9/9) (Figure 4(2)). The grafted cells projected to the striatum (N = 3/9 mice) and the hippocampus (N = 2/9 mice), known targets of the somatosensory cortex (Figure 4(5,8)) [39,40]. The most ventral donor axons observed were at the lateral amygdala (N = 1/9 mice), another known target of the somatosensory cortex (Figure 4(7)) [41]. The donor axons projected at least as far as 1 mm (the furthest distance examined) in both anterior (N = 2/9 mice) and posterior directions (N = 2/9 mice) at this 2 wpt timepoint (Figure 4(5,6)). We did not observe projections in areas the axons were not expected, such as the thalamus (Figure 4(9)). Overall, this demonstrates that donor neurons reached appropriate targets already within 2 wpt in the mature host cortex (Figure 1). 

### 3.6. Transplanted Neurons Become Electrophysiologically Active and Respond to Visual Stimuli

To test whether the donor neurons were functionally integrated in cortical networks and were able to fire action potentials, integrate with the host, and respond to external inputs such as sensory stimuli, we transplanted cells into the visual cortex (V1) of mice. An ultra-flexible neural probe designed for long-term recordings was inserted in the scaffold-cell mix at the time of transplantation and into the contralateral V1 that was intact and served as a control (Figure 5A). First, to observe the maturation of the donor neurons, we recorded non-evoked activity from freely moving mice starting 2–3 days after transplantation for 30 min sessions, 3 times a week for 12 weeks. At 2 wpt, there was very little activity. By 3 wpt, there was an increase in spikes and local field potential (LFP) consistent with the emergence of and increase in synaptogenesis (Figure 5B). One month after transplantation the “noise” subsided and the magnitude of the action potentials increased, possibly indicative of pruned synapses and maturing cortical tissue. Finally, at 5 wpt, higher amplitude spikes at a more regular frequency were observed, consistent with stable network activity. At 9 and 13 wpt, graft local field potential (LFP) power spectra are similar to the control. The corresponding LFP traces confirm the power spectra. Thus, the grafted neurons in this transplant model were maturing along a normal schedule and fired spontaneous action potentials (Luhmann et al., 2016).

To determine whether the donor neurons integrated into the visual cortical network and responded to external visual stimuli, we recorded from mice that were head fixed and presented visual stimuli once a week from 1–13 wpt (Figure 6A). The visual stimulus program displayed bars in six orientations and moving in two directions (right to left and left to right). Recorded units were classified as regular spiking cells (RS), which are putative excitatory neurons, and fast spiking cells (FS), which are putative inhibitory interneurons. Examples of waveforms of regular spiking cells corresponding to putative excitatory neurons in the control (spike width = 1.10 ms, FR = 2.3 HZ) and graft (spike width = 1.10 ms, FR = 2.3 HZ) are shown in Figure 6B. Neurons in the graft became visually responsive as early at 2 wpt (signal to noise ratio, SNR = 0.175, *p* = 0.00002, *n* = 2/3); however, at this early graft age, the responses of transplant-derived cells were predominantly present at the stimulus offset (Figure 6C). At 4 wpt, there was evidence of neuronal maturation. For example, a donor neuron displayed increased signal to noise ratio (SNR = 0.303, *p* < 0.001) and fired at stimulus onset, consistent with maturation. However, not all neurons appeared to be maturing at the same rate since a different neuron at 4 wpt continued to display a stimulus offset response (1 s delay) with a lower SNR (SNR = 0.234, *p* = 0.001) (right panel).

At 8 wpt, the firing rate was topically increased at stimulus onset and sustained for 1 s, followed by 1.5 s of a low firing rate (SNR = 0.919, *p* < 0.001). Finally, at 13 wpt, we consistently observed SNR comparable to that of units on the control side (SNR = 0.944, *p* < 0.001), which included donor neurons firing sparsely and being highly driven by the stimulus onset, consistent with donor cells maturing into functional adult pyramidal neurons. The orientation selectivity index (OSI) of the donor neurons increased between 2 (OSI = 0.23) and 4 wpt (OSI = 0.25) (Figure 6D), indicating the graft was becoming more tuned to specific orientations. These data provide evidence that the donor neurons in the primary visual cortex are synaptically integrating with the host and processing external visual stimuli, potentially via thalamic input, as would a normal developing V1 cortex [42]. 

## 4. Discussion

As a platform to rebuild neocortical tissue in vivo, we created a cortical transplant model that can support multiple cortical cell types, is vascularized, and is amenable to layering. The majority of cortical transplantation studies rely on the injection of cells directly into the parenchyma, such as through a Hamilton syringe or pulled glass pipette [2,43,44]. Such methods provide the experimenter with little control over the exact shape of the graft or arrangement of the donor cells within it, resulting in disorganized and unpredictable tissue structures.

To generate consistently sized lesions and increase transplant precision, we used an aspiration lesion. In this study, we removed healthy host cortical tissue, but in the future a similar approach may be used to remove degenerated or scarred tissue. The space created by the aspiration lesion then allows for the grafting of donor cells with a scaffold to provide more control over the final overall shape and size of the resulting tissue. Scaffolds and signaling factors within them can also increase donor cell survival and differentiation [21,45].

In addition, our transplant model provides a platform for organizing cells into layers. The mature neocortex has a distinct six-layered structure, with each layer containing specific combinations of neuronal subtypes. While cortical transplantation studies show that subtypes of neurons differentiate in grafts, they are not laminated as in the mature cortex [6,46]. A lack of normal lamination could prevent proper wiring of the donor neurons both with each other and with the host cortex. The method described here allows for the layering of precursor cells, which should in the future provide better control in achieving mature tissue that more closely recapitulates a normal neocortex. The goal would not be to deposit in the lesion site the neurons themselves to form the six-layered neocortex, but instead to deposit, for example, the layering of the different cell types that form the developing ventricular zone, subventricular zone, subplate, and marginal zone [46]. This could increase the likelihood of normal development of the transplant and better integration into the host cortex, potentially allowing for true cortical repair.

Another benefit of the grafting platform described here is that new tissue is assembled in vivo, which facilitates its rapid vascularization. Nutrients and oxygen can only diffuse ~400 um, necessitating any aerobic tissue larger than 400 um to establish and maintain a vascular system [47]. Vascularization is often a major hurdle in the expansion and long-term survival of 3D cultures and bioengineered tissues [47]. Proper vascularization of brain tissue is more of a challenge because it is highly metabolic, requiring 20% of the total oxygen consumed by a human [48]. Vessels in the graft are observed as early as 2 dpt, with significant vascularization occurring by 7 dpt. Neurogenic niches in mammals have lower levels of oxygen compared to more mature tissue, so the short delay in vascularization immediately post-transplant may not be an issue [49].

Despite the presence of vascular endothelial cells in our donor cell population and their importance in contributing new functional blood vessels after transplantation at the site of ischemic strokes [24], the vessels observed here after transplantation to the sites of aspiration lesions are primarily host-derived. Nevertheless, we show that the vessels in these grafts have circulating RBCs and are, therefore, likely transporting oxygen and nutrients to the new tissue to promote survival and integration. 

A requirement for functionally replacing neocortical tissue is that transplant-derived neurons can functionally integrate with the host brain. Using our paradigm, neurons were found to differentiate and project to appropriate targets in the host brain. Moreover, transplant-derived neurons were spontaneously active, integrated to the extents examined into functional networks, and responded to sensory inputs when grafted into V1. In the future, grafting into other parts of the neocortex will need to be tested as well. Nevertheless, these findings demonstrate that our transplant model supports functional neurons.

Current human disease modeling mostly relies on in vitro models derived from human pluripotent stem cells, which are often simple in their cell type composition and variable in their structure, in addition to being constrained by short survival due to the absence of vascularization and blood circulation [50]. Therefore, in vivo models to study human tissues would be advantageous. Interspecies chimeric models are becoming increasingly accepted as a method to investigate human diseases [44,51]. Our study suggests that the adult neocortex of immune compromised mice could be used as a platform for studying the cellular and molecular interactions between human neurons, vascular cells, and glia, and serve as a model for studying human neocortical disease, drug testing, and therapeutic tissue replacement.

## Figures and Tables

**Figure 1 bioengineering-10-00263-f001:**
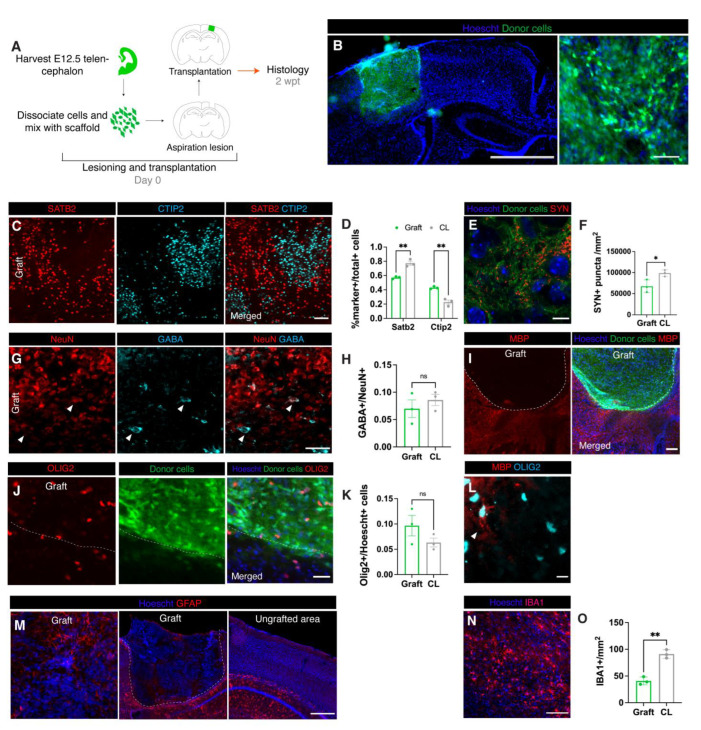
Transplanted embryonic cells in matrix differentiate at the site of aspiration lesions. (**A**)**:** Experimental design. (**B**)**:** Representative immunofluorescence image of transplant at 2 wpt at low (left; scale bar = 1550 μM) and high (right; scale bar = 100 μM) magnification. (**C**)**:** Representative immunofluorescence image of upper (SATB2) and deeper (CTIP2) cortical neurons in a graft (scale bar = 100 μM). (**D**)**:** Proportion of cells positive for cortical layer markers (SATB2: graft 57% of total SATB2 and CTIP2-labeled cells, N = 3; contralateral cortex 77%, N = 3, *p* = 0.003; CTIP2: graft 43%, N = 3; contralateral cortex 23% *p* = 0.003). (**E**)**:** Representative immunofluorescence image of anti-SYN staining (synapses) in a graft (scale bar = 5 μM). (**F**)**:** Density of anti-SYN fluorescence (graft 67,610 SYN+ puncta/mm^2^, N = 3; contralateral cortex 98,886 SYN+ puncta/mm^2^, N = 3, *p* = 0.03). (**G**)**:** Representative immunofluorescence image of mature neurons and inhibitory neurons (scale bar = 50 μM). Triangles indicate NeuN+/GABA+ cells. (**H**)**:** Quantification of inhibitory neurons (GABA) out of total mature neurons (NeuN) (graft: 7% GABA+ out of total NeuN+ cells, N = 3; contralateral cortex, 9% GABA+, N = 3, *p* = 0.45). (**I**)**:** Representative immunofluorescence image of myelin along the graft host border (dotted line) (scale bar = 200 μM). (**J**)**:** Representative immunofluorescence image of OLIG2+ oligodendrocyte lineage cells along the graft host border (dotted line) (scale bar= 50 μM). (**K**)**:** Quantification of OLIG2+/Hoescht+ cells (graft 9.7%, N = 3, contralateral cortex 6.3% N = 3, *p* = 0.2). (**L**)**:** Representative immunofluorescence image of a rare myelin (MBP)-positive oligodendrocyte in a graft (scale = 20 μM). Triangle indicates a myelinated OLIG2+ cell. (**M**)**:** Representative immunofluorescence images of astrocytes in a graft and contralateral cortex. Dotted line indicates outline of graft (scale bar = 500 μM). (**N**)**:** Representative immunofluorescence image of IBA1+ cells in a graft (scale bar = 100 μM). (**O**)**:** Density of IBA1+ cells (graft 40.8/mm2 N = 3, contralateral cortex 91.1/mm2, *p* = 0.21) *, *p* ≤ 0.05; **, *p* ≤ 0.01.

**Figure 2 bioengineering-10-00263-f002:**
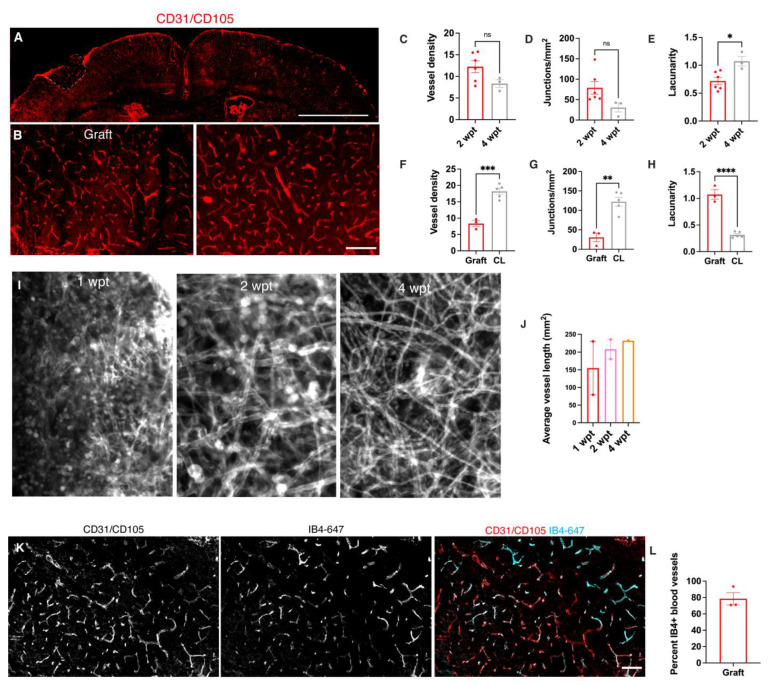
Grafts become functionally vascularized. (**A**)**:** Representative immunofluorescence image of vascularized cortex with graft (dotted line) at 2 wpt (scale bar = 2 mm). (**B**)**:** High magnification images of graft and contralateral cortex (scale bar = 200 μM). (**C**–**E**). Analysis of vasculature between grafts at 2 and 4 wpt. (**C**)**:** Blood vessel density (2 wpt 12.2% vessel area over total area, N = 6; 4 wpt 8.3%, N = 3, *p* = 0.1). (**D**)**:** Junction and branch point density (2 wpt, 79/mm^2^, N = 6; 4 wpt, 30.1/mm^2^, N = 3, *p* = 0.08). (**E**)**:** Lacunarity (2 wpt, 0.72, N = 6; 4 wpt, 1.1, N = 3, *p* = 0.02). (**F–H**). Analysis of vasculature between 4 wpt grafts and contralateral cortex. (**F**)**:** Blood vessel density (graft, 8.3% vessel area over total area, N = 3; contralateral cortex 18.2 %, N = 5, *p* = 0.0004). (**G**)**:** Junction and branch point density (graft 30.6/mm^2^, N = 3; contralateral cortex 122.4 N = 5, *p* = 0.02). (**H**)**:** Lacunarity (graft lacunarity = 1.1, N =3; contralateral cortex 0.3, N = 5, *p* < 0.0001). (**I**)**:** Representative 2-photon images of flattened Z-stacks of grafts at 1, 2, and 4 wpt. (**J**)**:** Average vessel length (1 wpt 154.2 mm^2^, N = 2; 2 wpt 208.2 mm^2^, N = 2; 4 wpt 232.4 mm^2^, N = 1). (**K**)**:** Representative immunofluorescence image after systemic perfusion with IB4 in a graft co-stained with total vessels (Scale bar = 200 μM). (**L**)**:** Percentage of IB4+ vessels in grafts normalized to total vessels (78.5 % IB4+ vessels, N = 3). *, *p* ≤ 0.05; **, *p* ≤ 0.01; ***, *p* ≤ 0.001; ****, *p* ≤ 0.0001.

**Figure 3 bioengineering-10-00263-f003:**
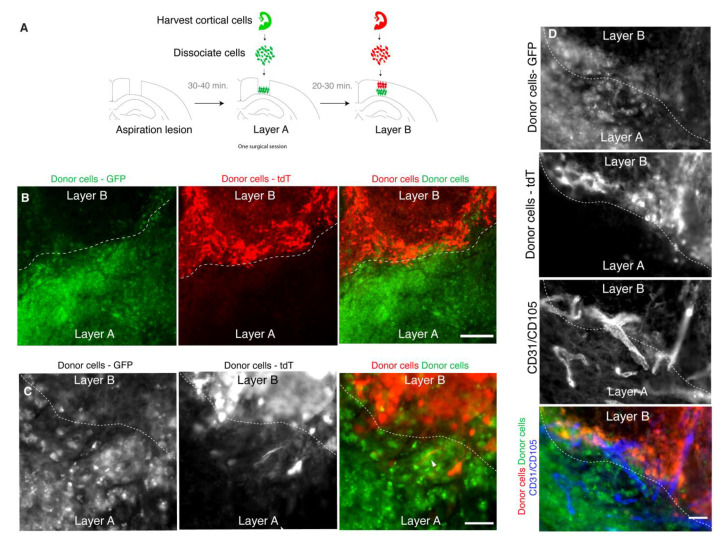
Grafts can be constructed in layers. (**A**)**:** Experimental design. (**B**)**:** Representative immunofluorescence image of a layered transplant at 2 wpt (scale bar = 200 μM). Dotted line indicates border between the layers. (**C**)**:** Example of a neuronal projection (arrowhead) crossing the border between layers (scale bar = 50 μM) already at 2 wpt. (**D**)**:** Example of blood vessels that cross the border between layers (scale bar = 50 μM).

**Figure 4 bioengineering-10-00263-f004:**
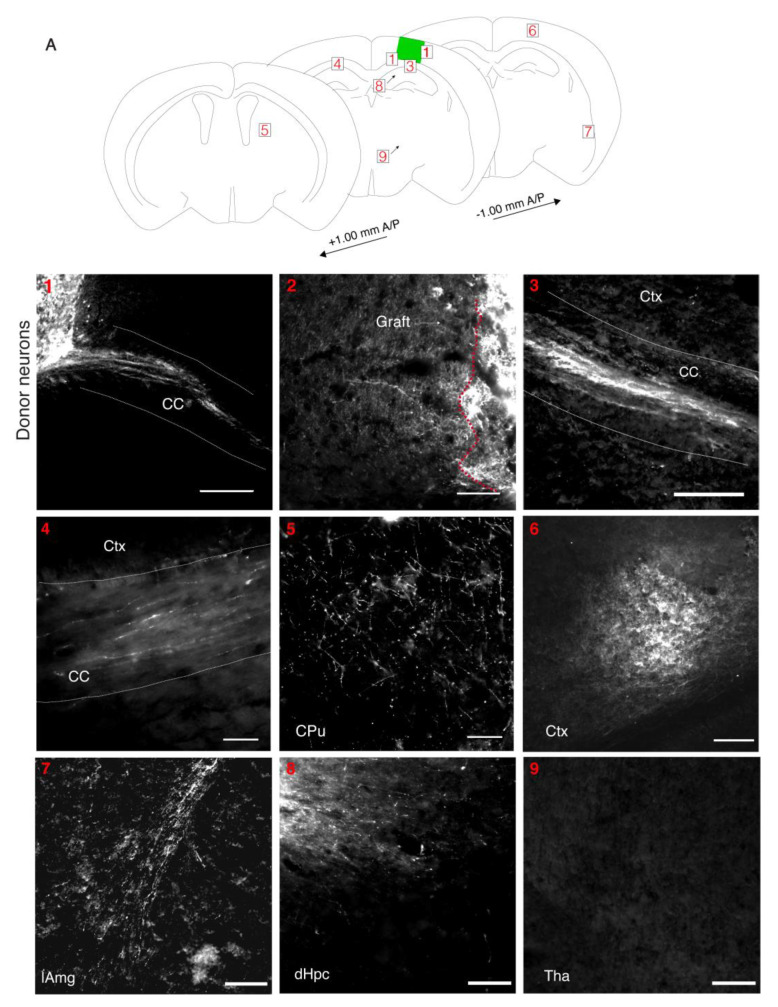
Mouse donor cells project to appropriate brain regions. (**A**)**:** Diagram of areas in host brain that donor cells project to at 2 wpt. Projections are seen in (**1**). The corpus callosum (CC) exiting the graft (N = 6/9) (scale bar = 300 μM). (**2**). The cortex directly adjacent to the graft (N = 9/9) (scale bar = 80 μM). (**3**). The CC away from the graft towards the contralateral hemisphere (N = 6/9) (scale bar = 80 μM). (**4**). The CC in contralateral cortex (N = 6/9) (Scale bar = 40 μM). (**5**). The caudate putamen (CPu) (N = 3/9) (scale bar = 80 μM). (**6**). The posterior cortex (N = 2/9) (scale bar = 100 μM). (**7**). The lateral amygdala (lAmg) (N = 1/9) (scale bar = 200 μM). (**8**). The dorsal hippocampus (dHPC) (N = 2/9) (scale bar = 100 μM). (**9**). The thalamus (Thal) (N = 9/9) (scale bar = 100 μM).

**Figure 5 bioengineering-10-00263-f005:**
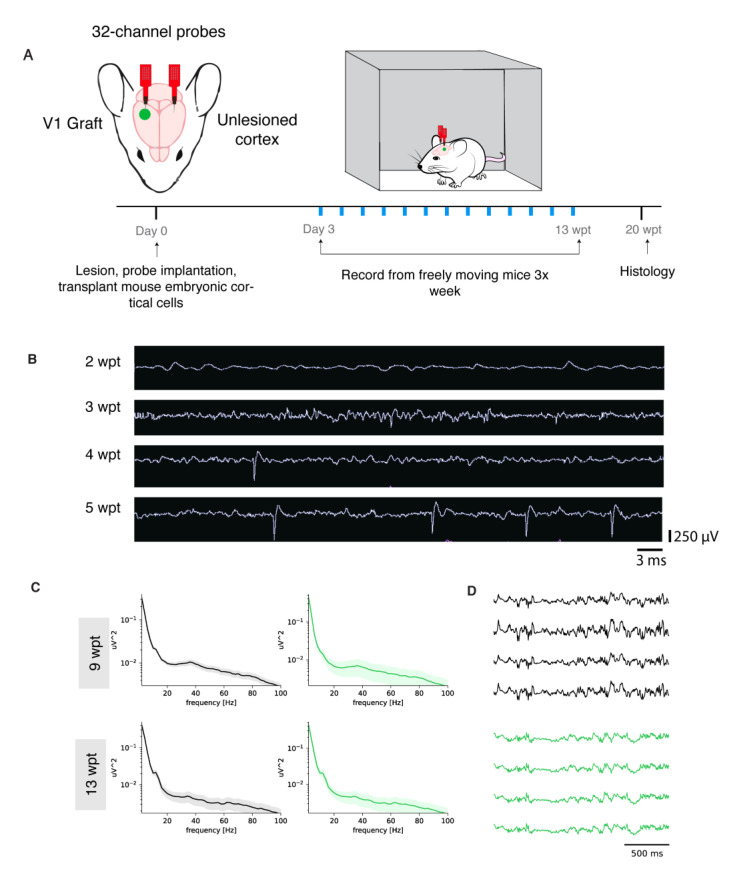
Mouse donor neurons become physiologically active. (**A**)**:** Experimental design. (**B**)**:** Neural traces of a single representative channel at 2, 3, 4, and 5 wpt with the same time and voltage scales. (**C**)**:** LFP power spectrums of control (left panels) and graft (right panels) at 9 and 13 wpt. (**D**)**:** LFP traces of control (gray) and graft (green) at 9 and 13 wpt.

**Figure 6 bioengineering-10-00263-f006:**
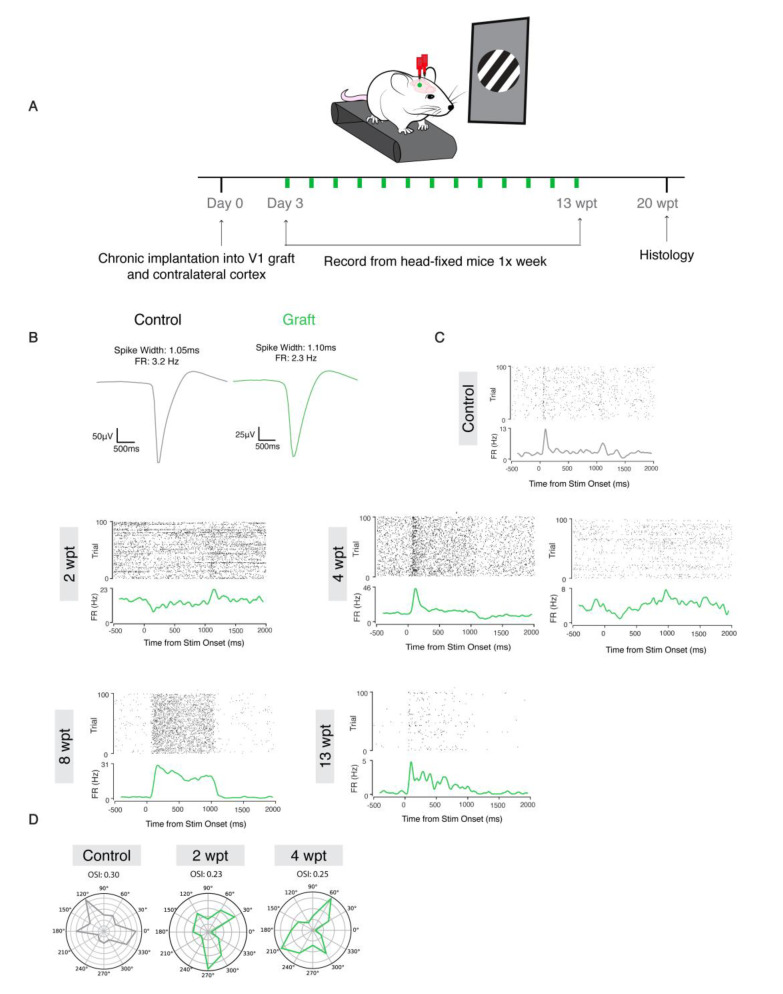
Mouse donor neurons respond to visual stimuli and are tuned to specific orientations. (**A**)**:** Experimental design. (**B**)**:** Representative waveforms of putative neurons from control (left:black) and graft (right:green). (**C**)**:** Raster plots of firing rate and signal to noise ratio (SNR) at 2, 4, 8, and 13 wpt (from left to right, top to bottom). (**D**)**:** OSI for control and graft at 2 and 4 wpt.

**Table 1 bioengineering-10-00263-t001:** **Number of mice used for each experiment listed by figure**. Asterisk denotes that in these mice the intact contralateral side was used as the control.

Figure	1D	1F	1H	1K	1O	2C-E	2F-H	2I,J	2L	
**Number of host mice**	3 *	3 *	3 *	3 *	3 *	9	8 *	5	3	
**Figure**	3	4	5	6	S1A	S1B	S1C	S2C,D	S2E	** Total**
**Number of host mice**	2	9	2 *	3 *	3	80	6	19	10	**200**

* Note that in these mice the intact contralateral side was used as the control.

**Table 2 bioengineering-10-00263-t002:** **Antibody list**. Asterisks indicate use of antigen retrieval.

Table	Company	Catalog No.	Species	Dilution
Anti-CD105	Biolegend	120402	Rat	1:50 *
Anti-CD31	BD Pharmingen	553370	Rat	1:50 *
Anti-CTIP2	abcam	ab18465	Rat	1:500 *
Anti-GABA	Millipore-Sigma	ab2052	Rabbit	1:1000
Anti-GFAP	Invitrogen	13-0300	Rat	1:500
Anti-GFP	ThermoFisher	A11122	Rabbit	1:250
Hoechst 33342	Invitrogen	H3570		1:1000
IB4-647	ThermoFisher	I32450		1 μg/μL
Anti-Iba1	Wako	019-19741	Rabbit	1:100
Anti-MBP	abcam	ab7349	Rat	1:50
Anti-NeuN	Synaptic Systems	266004	Guinea Pig	1:500
Anti-OLIG2	Millipore-Sigma	ab9610	Rabbit	1:50
Anti-SATB2	abcam	ab92446	Rabbit	1:500 *
Anti-Synaptophysin	abcam	ab32127	Rabbit	1:500 *

## Data Availability

Not Applicable.

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
