# Peer review of "An In Vivo Platform for Rebuilding Functional Neocortical Tissue"

_bioengineering, 2023, doi:10.3390/bioengineering10020263_

Round 1

Reviewer 1 Report

Dear Dr. Jean,

I am pleased to review the article entitled: An in vivo platform for rebuilding functional neocortical tissue. I found this study quite interesting and useful for the application of grafts. the disease modeling area, this is very benefiting to apply this kind of approach. However, few minor corrections are suggested in the following:

1.Line 44-45: This is unsurprising considering….. l cortical cytoarchitecture. This sentence need to rephrase as it seems unclear.

2. Please check reference format within the text and in reference section, if this is right or need to modify.

3. Line 63: there is an extra space in the sentence …… for the experiment

 4. The method section, in animals detail, the number of animal in each group is not mentioned. the alternative suggestion for the detail of each transgenic group, it is better to prepare a flow chart or table with the detail of each group and cross of bread for each type of mice group with the number of animal per group used in this study.

Best regards,

Author Response

We were pleased that the Reviewers were overall positive and thankful for their careful reading and consideration of the manuscript. We have now uploaded a revised manuscript. In the new version, we have made edits to the text (in blue) that address the reviewers’ comments and that we believe have made the manuscript stronger.

Reviewer 1 comments and responses:

I am pleased to review the article entitled: An in vivo platform for rebuilding functional neocortical tissue. I found this study quite interesting and useful for the application of grafts. the disease modeling area, this is very benefiting to apply this kind of approach. However, few minor corrections are suggested in the following:

1.Line 44-45: This is unsurprising considering….. l cortical cytoarchitecture. This sentence need to rephrase as it seems unclear.

Response: The sentence has been clarified, and now reads: “The inability to demonstrate that electrophysiological activity of grafted neurons encode useful behavior is not surprising considering there are cortical cell types that are thus far missing in grafts, in addition to these grafts lacking normal cortical cytoarchitecture.”

  1. Please check reference format within the text and in reference section, if this is right or need to modify.

Response: Citations in the text and references were checked, and typos and misformating corrected.

  1. Line 63: there is an extra space in the sentence …… for the experiment

Response: The mistake has been corrected.

  1. The method section, in animals detail, the number of animal in each group is not mentioned. the alternative suggestion for the detail of each transgenic group, it is better to prepare a flow chart or table with the detail of each group and cross of bread for each type of mice group with the number of animal per group used in this study.

Response: We have now added a table in the Methods describing the number of mice for each experiment and their total.

Reviewer 2 Report

In the manuscript titled "An in vivo platform for rebuilding functional neocortical tissue", the authors performed small lesion by aspirating the neocortical tissue and injecting a mixture of embryonic telencephalic cells resuspended in matrigel. 2 weeks post lesion, the grafted cells had differentiated into the main neural cell types (including excitatory neurones, oligodendrocytes, and interneurones) albeit, not always in the expected ratio. The grafted matrix was permissible to the growth of vessels, allowing vascularisation. Critically, the authors were able to demonstrate restoration of connectivity and neural activity.

Overall, the study was well designed and provides exciting prospects for brain repair following trauma. The figures are clearly laid out and the results well presented.

Some aspects of the study need to be clarified. Why was the embryonic stage E12.5 chosen for the graft?

This early stage of corticogenesis corresponds to the lamination of infragranular neurons, typically CTIP2+. This is confirmed by the over-representation of CTIP2+ neurons in the grafted tissue.

Have the authors attempted their approach with E14.5 donor cells? This might rebalance the supragranular vs. infragranular ratio compared to the control tissue, or potentially a combination of E12-14.5 donor cells.

The authors demonstrate that they are able to deposit successive layers of cell/Matrigel mixture, which didn’t allow cell movement across the layer boundary. They suggest this would be a way to artificially replicate the cortex laminar structure. However, the primate, including human, neocortex is comprised of a much more complex laminar organisation. It is therefore unrealistic to expect to reproduce this organisation, especially since it is also cortical area dependent. Have the authors considered a way of improving the Matrigel environment to promote cell movement and differentiation to better mimic the cellular composition and organisation of the surrounding native tissue? The use of nanomaterial to which growth factors can be tethered to create distinct microenvironments would be a more viable approach for future therapeutic applications.

The cellular composition was analysed at 2 weeks post-surgery. Have the authors performed a similar analysis at later timepoints, particularly at 13wpt when the electrophys experiments reveal that the neural activity in the graft correlates with that of the uninjured cortex.

Finally, the authors have not performed negative controls, in which the surgery was performed but no cells were present in the Matrigel. These are critical to confirm the importance of the cells in the restoration of the neural activity rather than just an effect of a clean tissue ablation and rapid filling with the Matrigel which is conducive to revascularisation and prevents glial scarring which is the greatest impediment to the recovery of neural function following lesion.

Minor comments:

There are a few typos in the manuscript. 

Line 448: replace "tegular" by "regular"

Line 451: replace "ration" by "ratio"

Line 464: replace "topically" by "typically"

Figure 1: the legend indicates the scale bar length in µM (which is micromolar) instead of µm which are micrometer, please correct accordingly. 

Author Response

We were pleased that the Reviewers were overall positive and thankful for their careful reading and consideration of the manuscript. We have now uploaded a revised manuscript. In the new version, we have made edits to the text (in blue) that address the reviewers’ comments and that we believe have made the manuscript stronger.

Reviewer 2 comments and responses:

In the manuscript titled "An in vivo platform for rebuilding functional neocortical tissue", the authors performed small lesion by aspirating the neocortical tissue and injecting a mixture of embryonic telencephalic cells resuspended in matrigel. 2 weeks post lesion, the grafted cells had differentiated into the main neural cell types (including excitatory neurones, oligodendrocytes, and interneurones) albeit, not always in the expected ratio. The grafted matrix was permissible to the growth of vessels, allowing vascularisation. Critically, the authors were able to demonstrate restoration of connectivity and neural activity.

Overall, the study was well designed and provides exciting prospects for brain repair following trauma. The figures are clearly laid out and the results well presented.

Some aspects of the study need to be clarified. Why was the embryonic stage E12.5 chosen for the graft?

Response: We have added a sentence to the 1st paragraph of the 2nd section of the Results, providing an additional rationale supporting the use of E12.5 embryos as a source of cerebral precursors: “In addition, we previously observed that E12.5 cortex-derived vascular, glial, and neuronal precursor cells survive, differentiate, and integrate with the adult mouse cortex (Krzyspiak et al., 2022)”.

This early stage of corticogenesis corresponds to the lamination of infragranular neurons, typically CTIP2+. This is confirmed by the over-representation of CTIP2+ neurons in the grafted tissue.

Have the authors attempted their approach with E14.5 donor cells? This might rebalance the supragranular vs. infragranular ratio compared to the control tissue, or potentially a combination of E12-14.5 donor cells.

Response: We agree, changing the age of the neocortical precursors is expected to change the ratio of the layer types obtained. However, our goal in this study was not to obtain normal ratios of neocortical cell types (or their correct lamination), but to establish that the lesion and cell/matrix grafting approach described here could be used for doing so in the future by showing that the subtypes can survive and differentiate (and that cells can be deposited in the graft site in layers). See also response below regarding obtaining layers.

The authors demonstrate that they are able to deposit successive layers of cell/Matrigel mixture, which didn’t allow cell movement across the layer boundary. They suggest this would be a way to artificially replicate the cortex laminar structure. However, the primate, including human, neocortex is comprised of a much more complex laminar organisation. It is therefore unrealistic to expect to reproduce this organisation, especially since it is also cortical area dependent. Have the authors considered a way of improving the Matrigel environment to promote cell movement and differentiation to better mimic the cellular composition and organisation of the surrounding native tissue? The use of nanomaterial to which growth factors can be tethered to create distinct microenvironments would be a more viable approach for future therapeutic applications.

Response: To clarify, the goal moving forward is not to artificially replicate the laminated cortical structure by depositing these layers themselves. Instead, what layering would allow is the initial organization of early fetal-like cell layers (e.g. VZ and preplate for the earliest stages, or VZ, SVZ, nascent CP, MZ for later stages), which would then generate the more mature layers themselves. Note, we already know from our previous work (Krzyspiak et al., 2022, Stem Cell Research) and from several organoid studies that cortical precursors are poised and capable of generating, at least to some extent, layers on their own. So our goal would be to enhance the fidelity and normalcy of this process. We have added text to the 3rd paragraph of the Discussion to clarify this: “The goal would not be to deposit in the lesion site the neurons themselves to form the 6-layered neocortex, but instead to deposit…”. Similarly, regarding the extracellular composition of early layers, we have a separate project aimed at characterizing major scaffolding proteins, complex carbohydrates, and growth factors to approximate a more normal environment.

The cellular composition was analysed at 2 weeks post-surgery. Have the authors performed a similar analysis at later timepoints, particularly at 13wpt when the electrophys experiments reveal that the neural activity in the graft correlates with that of the uninjured cortex.

Response: Unfortunately, we were unable to because removal of the electrode/head implant at the time of euthanasia would rip out the grafts, leaving us unable to perform histology at the end of the experiment. Future experiments should include histology on separate cohorts of 13 wpt grafts that do not have the head implants to correlate the cytoarchitecture, cell composition, and circuitry of the graft with its electrophysiology.

Finally, the authors have not performed negative controls, in which the surgery was performed but no cells were present in the Matrigel. These are critical to confirm the importance of the cells in the restoration of the neural activity rather than just an effect of a clean tissue ablation and rapid filling with the Matrigel which is conducive to revascularisation and prevents glial scarring which is the greatest impediment to the recovery of neural function following lesion.

Response: Matrigel does have beneficial properties. However, given the lack of neurogenesis in the adult neocortex, it would seem unprecedented that new host-derived cortical neurons would colonize the Matrigel in a cell-free Matrigel-only graft and generate the observed neuronal maturation signatures and responses to visual stimuli. And it also seems unlikely that in the experiments reported here with donor cells in Matrigel that new host neurons would infiltrate the graft to generate maturing neural activity since all neurons we have observed in these grafts are donor-derived. Therefore, a Matrigel only control did not seem necessary.

Minor comments:

There are a few typos in the manuscript. 

Line 448: replace "tegular" by "regular"

Response: Corrected.

Line 451: replace "ration" by "ratio"

Response: Corrected.

Line 464: replace "topically" by "typically"

Response: Corrected.

Figure 1: the legend indicates the scale bar length in µM (which is micromolar) instead of µm which are micrometer, please correct accordingly. 

Response: Corrected in Figure 1, as well as in other figures.

Reviewer 3 Report

The manuscript entitled " An in vivo platform for rebuilding functional neocortical tissue” is an experimental manuscript where the authors have transplanted embryonic cells to recreate the cortical layers of cells in vivo. The authors have performed various experiments in a logical manner to analyze the study outcome, there is high novelty to the study. However, the authors have missed few concepts and some major concerns are described below.

1.       More elaborate introduction can be written and please mention clearly the goal of the study.

2.       Methods can be made more clear, for example – it was not very clear if the injection was unilateral or bilateral. How many cells were injected? how many cells per layer? what are the number of animals and groups used in the study?

3.       There are some methods discussed in the results section, please add them to the methods section.

4.       Though the authors have used staining techniques to validate the layers, it is not very clear if all the cell types in the 6 layers of the cortex were analyzed.

5.       What was the purpose of analyzing the visual cortex which is the layer IV. How about the functions aspects from other layers.

6.       In embryos, the cortical layers mature inside out, deep layers mature first, Is there any evidence of the maturation pattern in this platform analyzed compared to the what happens during the embryogenesis.

7.       The visual testing and the correlation to the V1 firing response are a weakly designed experiment. Would it be possible that the firing in the in V1 and the animals responding to the visual stimulus could be due to brain plasticity/ocular dominance that the intact V1 in the contralateral side of the hemisphere is compensating for the lesioned hemisphere? The authors have only shown vertical bars/lines which are processed in V1. However, to make sure that the V1 makes connections with V2 in the lesioned hemisphere and the newly formed cells can make connections, complex visual tasks should be performed.

8.       If according to the authors, the newly formed neurons are able to make appropriate connections, more evidence, especially behavioral evidence should be given as a confirmation.

9.       It would also be beneficial to analyze the neurotransmitter release in the brain following the grafting experiment.

Conclusion: Though the study results are encouraging, the authors have not given solid evidence of all 6 layers being regenerated as well as behavioral and functional evidence to justify this. The manuscript should be reorganized, and the goal clarified. As of now, the manuscript results and conclusions are weak and cannot be accepted in this form. This reviewer recommends rejection of this manuscript.

Author Response

We were pleased that the Reviewers were overall positive and thankful for their careful reading and consideration of the manuscript. We have now uploaded a revised manuscript. In the new version, we have made edits to the text (in blue) that address the reviewers’ comments and that we believe have made the manuscript stronger.

Reviewer 3 comments and responses:

The manuscript entitled " An in vivo platform for rebuilding functional neocortical tissue” is an experimental manuscript where the authors have transplanted embryonic cells to recreate the cortical layers of cells in vivo. The authors have performed various experiments in a logical manner to analyze the study outcome, there is high novelty to the study. However, the authors have missed few concepts and some major concerns are described below.

  1. More elaborate introduction can be written and please mention clearly the goal of the study.

Response: We have added the following sentence to the last paragraph of the Introduction to elaborate on the goal of the study: “The goal of this study is to provide an initial proof-of-concept for a neocortical grafting platform that supports 1) the survival and differentiation of the major neocortical cell types, 2) vascularization, 3) neuronal integration, and 4) layering.”

  1. Methods can be made more clear, for example – it was not very clear if the injection was unilateral or bilateral. How many cells were injected? how many cells per layer? what are the number of animals and groups used in the study?

Response: In all cases, cells and matrix were applied unilaterally, only on the side that had the lesion. We have clarified this in several parts of the manuscript (in the 1st sentence of the Results, in the 1st paragraph of the second section of the Results, and in the Methods section “Transplantation procedures”). The number of cells injected for the 1.25 diameter lesions used in the study is roughly 750K. The number of cells in layered grafts were similar (~375K cells for each layer). We have now stated this in the Methods section “Transplantation procedures”. For the number of animals, we now provide a Table as described in our response to Reviewer 1’s point 4.

  1. There are some methods discussed in the results section, please add them to the methods section.

Response: We have now added details regarding certain experimental approaches (previously only described in the Results) to the Methods section, including to “In vivo live imaging”, Visual stimulation”, and a new short section “In vivo freely-moving electrophysiological recordings”.

  1. Though the authors have used staining techniques to validate the layers, it is not very clear if all the cell types in the 6 layers of the cortex were analyzed.

Response: For this study, we were interested in determining whether this transplant model supports the survival of the major cell types, including for the excitatory neurons two large subclasses, supragranular and infragranular neurons, without doing an in depth analysis of all subtypes (this was also the case for the inhibitory neurons). In future studies, high resolution single cell sequencing will be used to answer this question.

  1. What was the purpose of analyzing the visual cortex which is the layer IV. How about the functions aspects from other layers.

Response: Area V1 of the neocortex, for which all layers 1-6 process information, has been extensively studied in mice, and tools are readily available to characterize the function and physiology of its neurons. Therefore, for this proof-of-concept study, we took advantage of the knowledge and tools available for V1. In future studies, it will be useful to also examine graft performance in other cortical areas. We have added text to this effect in the Discussion, 2nd to last paragraph: “…when grafted into V1. In the future, grafting into other parts of the neocortex will need to be tested as well. Nevertheless,…”.

  1. In embryos, the cortical layers mature inside out, deep layers mature first, Is there any evidence of the maturation pattern in this platform analyzed compared to the what happens during the embryogenesis.

Response: We did not examine whether infragranular neurons differentiate before supragranular ones in our grafts. This would be interesting to confirm in future studies. The strong expectation already, however, is that this would be the case given that stem cell-derived cortical precursors even in culture show a temporal progression from generating infragranular to supragranular neurons (reviewed for example by Ehsaei et al., 2018, Neuropsychi. 8:1715).

  1. The visual testing and the correlation to the V1 firing response are a weakly designed experiment. Would it be possible that the firing in the in V1 and the animals responding to the visual stimulus could be due to brain plasticity/ocular dominance that the intact V1 in the contralateral side of the hemisphere is compensating for the lesioned hemisphere? The authors have only shown vertical bars/lines which are processed in V1. However, to make sure that the V1 makes connections with V2 in the lesioned hemisphere and the newly formed cells can make connections, complex visual tasks should be performed.

Response: There are several factors that argue against involvement from the non-lesioned side. First, the firing that was recorded and labeled as “graft” was with an electrode in the graft at the lesion site, which only has donor but not host neurons, and which only detects input from neurons <100 um away from each channel. Second, the monitors that displayed the visual stimuli were presented to only one eye at a time, making it very unlikely that the V1 ipsilateral to a stimulus triggered responses in the V1 contralateral to the stimulus that are coincident in time with that stimulation. Finally, the critical period for ocular dominance plasticity is closed at P30 in mice, and the host mice used here are >2 months old.  Future studies will investigate functional connectivity of donor neurons in primary cortices with higher order areas.  

  1. If according to the authors, the newly formed neurons are able to make appropriate connections, more evidence, especially behavioral evidence should be given as a confirmation.

Response: Although the graft-derived neurons can make appropriate connections, this is a far cry from suggesting that the tissue is functional enough to encode useful behavior – we do not think that the grafted tissue can do this, nor do we make this conclusion. On the contrary, we suggest that this is a useful platform that should facilitate development of better tissue in the future, which might then be worth testing for its ability to encode parts of behavioral circuits.

  1. It would also be beneficial to analyze the neurotransmitter release in the brain following the grafting experiment.

Response: Analyzing the neurotransmitter phenotypes and release for the different graft-derived neuronal cell types is another good suggestion moving forward, but currently beyond the scope of this initial study.

Conclusion: Though the study results are encouraging, the authors have not given solid evidence of all 6 layers being regenerated as well as behavioral and functional evidence to justify this. The manuscript should be reorganized, and the goal clarified. As of now, the manuscript results and conclusions are weak and cannot be accepted in this form. This reviewer recommends rejection of this manuscript.

Response: We have clarified the goals of the study in the last paragraph of the Introduction.

Round 2

Reviewer 2 Report

The authors have addressed the comments raised during the first round of review. The manuscript is ready for publication.

Reviewer 3 Report

The authors have answered the reviewer comments in future expectations, the authors have directed most of their response towards future directions, which does not justify their answers or clarify the previous comments from this reviewer.

 Because of this lack of scientific present evidence, the enthusiasm of the article remains very low and incomplete. The reviewer still rejects this manuscript in the present form.